# Observing Dynamic Conformational Changes within the Coiled-Coil Domain of Different Laminin Isoforms Using High-Speed Atomic Force Microscopy

**DOI:** 10.3390/ijms25041951

**Published:** 2024-02-06

**Authors:** Lucky Akter, Holger Flechsig, Arin Marchesi, Clemens M. Franz

**Affiliations:** 1WPI Nano Life Science Institute, Kanazawa University, Kanazawa 920-1167, Japan; lucky.akter@stu.kanazawa-u.ac.jp (L.A.); flechsig@staff.kanazawa-u.ac.jp (H.F.); a.marchesi@staff.univpm.it (A.M.); 2Department of Experimental and Clinical Medicine, Università Politecnica delle Marche, Via Tronto, 10/A Torrette di Ancona, 60126 Ancona, Italy

**Keywords:** HS-AFM, laminin-111, laminin-332, laminin-511, coiled-coil, extracellular matrix, molecular dynamics

## Abstract

Laminins are trimeric glycoproteins with important roles in cell-matrix adhesion and tissue organization. The laminin α, ß, and γ-chains have short N-terminal arms, while their C-termini are connected via a triple coiled-coil domain, giving the laminin molecule a well-characterized cross-shaped morphology as a result. The C-terminus of laminin alpha chains contains additional globular laminin G-like (LG) domains with important roles in mediating cell adhesion. Dynamic conformational changes of different laminin domains have been implicated in regulating laminin function, but so far have not been analyzed at the single-molecule level. High-speed atomic force microscopy (HS-AFM) is a unique tool for visualizing such dynamic conformational changes under physiological conditions at sub-second temporal resolution. After optimizing surface immobilization and imaging conditions, we characterized the ultrastructure of laminin-111 and laminin-332 using HS-AFM timelapse imaging. While laminin-111 features a stable S-shaped coiled-coil domain displaying little conformational rearrangement, laminin-332 coiled-coil domains undergo rapid switching between straight and bent conformations around a defined central molecular hinge. Complementing the experimental AFM data with AlphaFold-based coiled-coil structure prediction enabled us to pinpoint the position of the hinge region, as well as to identify potential molecular rearrangement processes permitting hinge flexibility. Coarse-grained molecular dynamics simulations provide further support for a spatially defined kinking mechanism in the laminin-332 coiled-coil domain. Finally, we observed the dynamic rearrangement of the C-terminal LG domains of laminin-111 and laminin-332, switching them between compact and open conformations. Thus, HS-AFM can directly visualize molecular rearrangement processes within different laminin isoforms and provide dynamic structural insight not available from other microscopy techniques.

## 1. Introduction

Laminins are large (≤900 kDa) basement membrane glycoproteins with vital roles in development, the maintenance of tissue organization, cell migration, differentiation, neurite outgrowth, embryogenesis, and angiogenesis [1,2,3,4,5,6,7]. All laminin isoforms are heterotrimers consisting of α-, β-, and γ-polypeptide chains. In mammals, 16 different laminin isoforms have been identified from a combination of five α-chains, three β-chains, and three γ-chains, named using a standardized notation, for instance, designating α5β1γ1 as laminin-511 [8]. The laminin α-, β-, and γ-chains have short N-terminal arms, while their C-termini are connected via a long triple coiled-coil domain formed by the intertwining of ~600 amino acid α-helical sections from each chain [9], giving the laminin molecule a characteristic cross-shaped morphology. The laminin short arms consist of a globular laminin N-terminal (LN) domain and globular laminin L4 and LF domains inserted into an array of rod-shaped laminin-type EGF-like (LE) domains [10]. The laminin α-chain contains five additional globular laminin G-like (LG) domains downstream of the coiled-coil region. Ionic interactions within the coiled-coil domain determine the specificity of laminin chain assembly [11], while the coiled-coil structure is further stabilized by a disulfide bond between the β- and γ-chains near their C-termini [12]. Laminins also contain binding sites for integrin adhesion receptors, including α1β1, α2β1, α6β1, α6β4, and α7β1 [13], and for dystroglycan [14], thereby providing important cellular anchor points within tissues.

An integral function of laminin is to provide a molecular scaffold for epithelial and endothelial monolayers on basement membranes, which are thin highly specialized extracellular matrix sheets. Within basement membranes, laminins form networks independently [15,16] and by binding to other basement membrane constituents, including collagen IV, entactin, and heparin sulfate proteoglycan to form large supramolecular complexes [17]. Laminin-111 (previously known as laminin-1) is expressed early in embryogenesis, has an integral function in basement membrane assembly and function, and is frequently used as an adhesive coating to enhance integrin-mediated cell attachment and growth in tissue culture applications. Laminin-332 (formerly laminin-5) plays an important role in anchoring keratinocytes to the underlying dermis in the dermal–epidermal junctional zone [18]. It interacts with integrin α6β4, α3β1, and α6β1, contributes to wound healing and tumor invasion through potent cell-migration-promoting activity, and promotes the formation of hemidesmosomes via integrin α6β4-mediated interactions. In contrast to the α1-chain of laminin-111, which features five LG domains arranged in a tandem repeat configuration (LG1-3, LG4-5), the proteolytic processing of the α3-chain leads to the cleavage of the C-terminal LG4-5 domains from the LG1-3 module in mature laminin-332 [19]. Additional enzymatic processing leads to the trimming of the N-terminus of the γ2 chain [20,21], while the β3 chain remains intact [22,23]. Moreover, two transcript variants of the α3 chain that differ in the length of the N-terminal region produce a shorter α3A and a longer α3B chain, giving rise to laminin-3A32 and -3B32 isoforms, respectively. As a result of proteolytical processing and transcriptional variation, laminin-3A32 thus features truncated short arms and a truncated LG domain region compared to laminin-111.

Overall, laminin-111 ultrastructure and domain organization is well established from rotary shadowing electron microscopy (EM) studies, revealing a rigid, asymmetric cross-shaped structure consisting of three short arms and one long arm [24]. In contrast, an extended rod-like structure has been observed for laminin-332 using rotary shadowing electron microscopy [25,26,27]. However, EM imaging requires sample dehydration and coating, and only generates snapshots of individual conformational states. An increasing number of crystal structures of laminin subregions are also available, including LN domains [28,29], the nidogen-binding site of laminin within the LE domain of the γ1 chain, [30], individual laminin LG modules [31,32], or the C-terminal coiled-coil region with adjacent LG1-3 domains (mini-E8 fragment) of laminin-111 [33] and -511 [34], but atomistic structures covering large portions or entire laminin isoforms are still missing [10]. Two pioneering atomic force microscopy (AFM) studies investigated laminin structure under physiological conditions and confirmed the cross-shaped organization of laminin, while also visualizing some dynamic motion of laminin-111 molecules [35,36]. However, the limited time resolution of conventional AFM (about one frame per minute or less) makes it difficult to resolve the dynamics of biomolecules, while the use of unmodified AFM tips only yielded comparatively low-resolution images lacking submolecular structural detail. Unlike conventional AFM, high-speed AFM (HS-AFM) can directly visualize structural changes in biological molecules with subsecond temporal and subnanometer spatial resolution [37,38,39,40], but so far it has not been employed to investigate laminin dynamics.

In this study, we applied HS-AFM imaging to study the structural dynamics of laminin-111 and laminin-332 under physiological conditions. Our results demonstrate that the laminin-332 coiled-coil domain displays highly dynamic flexing around a defined central hinge, while the laminin-111 coiled-coil domains remain in a comparatively stable S-shaped configuration. Furthermore, we detected structural fluctuations within the C-terminal LG domain cluster of laminin-111 and -332 between compact and open confirmations, which could play a role in regulating adhesion receptor binding. HS-AFM imaging can thus reveal isoform-specific conformational changes occurring within different laminin domains, providing novel insights into dynamic laminin structure and function.

## 2. Results

### 2.1. Optimization Surface Immobilization of Laminin-111 for HS-AFM Imaging

HS-AFM is unique in that it can directly visualize the dynamic processes of proteins and other biomolecules in liquid conditions with nanometer resolution. However, the selection of an appropriate substrate for sample immobilization is one of the key factors for the successful HS-AFM observation of the biological molecule. For the successful visualization of biomolecular processes using HS-AFM, the substrate should have suitable properties, such as an appropriate binding affinity for the sample to retain its physiological function and mediate the attachment of molecules in their desired orientation [41]. Various surfaces have been used for the HS-AFM imaging of biological molecules, such as bare or chemically modified mica, supported lipid bilayers (SLBs) formed on mica, or highly oriented pyrolytic graphite (HOPG) [42]. 

Freshly cleaved natural muscovite mica provides atomically flat surfaces over large areas, and it is a commonly used substrate for HS-AFM imaging. We first tested if cleaved mica surfaces would also provide a suitable substrate for laminin-111 imaging using HS-AFM. However, at physiological pH (7.5), human recombinant laminin-111 molecules only attached weakly, complicating the analysis of laminin-111 ultrastructure due to the excessive flexibility of its elongated arm structure (Figure 1A, Appendix A). Specifically, the laminin short and long arms displayed great structural fluctuations, while the LG domain attached more stably. Under physiological conditions, K^+^ ions located between the cleaved mica layer desorb from the surface, and the resulting negatively charged mica surface then facilitates the electrostatic adsorption of positively charged molecules. The predicted pI of laminin-111 is 5.3, giving it a negative net charge at pH 7.5, which could account for the weak surface attachment to the negatively charged mica surface. We therefore tested if laminin-111 could be more stably attached at low pH, when it displays a positive net charge. As expected, lowering the pH to 5 or 3.5 promoted the stable attachment of laminin-111, making it now possible to image laminin ultrastructure and domain organization in detail (Figure 1A, Appendix A). Conversely, elevating the pH of the immobilization/imaging solution to pH 9 or 10 further weakened laminin-111 surface immobilization. With a decreasing pH of the imaging solution, the laminin short arms become increasingly entangled, as shown in previous conventional AFM studies of laminin [35,36]. As a result, only a few molecules were found in the typical cruciform structure of laminin-111 at pH 3.5. Nevertheless, in some well-spread molecules, the laminin ultrastructure could be visualized with high resolution, and the domain organization of the short arms could be distinguished, including the globular LN, L4a, and L4b subdomains of the α1-chain, the LN and LF domains of the β-chain, and the LN and L4 domains of the γ-chain, although the β- and γ-chains could not be distinguished due to their similar domain organization and length (Figure 1C).

While lowering the pH of the imaging buffer enhanced laminin-111 immobilization and increased image quality, subjecting proteins to unphysiological pH levels may negatively affect protein function, structure, and stability. For instance, at low pH, histidine side chains may become protonated, potentially affecting the hydrogen bonding network regulating protein structure [43]. Moreover, an overly strong surface attachment may reduce protein flexibility and prevent dynamic molecular rearrangement processes associated with normal protein function. We therefore explored alternative physisorption methods for laminin-111 immobilization compatible with physiological pH. For this, the mica surface was functionalized using 3-aminopropyltriethoxysilane (APTES), which introduces NH_3_^+^ groups onto the mica surface at physiological pH [44] and thus reverses the mica surface charge from negative to positive. Three different APTES concentrations (1%, 0.1%, and 0.01%) were tested and laminin-111 was then imaged at a physiological pH of 7.5 (Figure 1B, Appendix A). Laminin-111 was well attached at all conditions, and individual laminin domains could be routinely recognized. However, substrate roughness increased with an increasing concentration of APTES. Moreover, at high APTES concentrations (1%), the surface became sticky, which sometimes led to AFM tip contamination and decreased image quality. On the other hand, at 0.01% APTES concentration, surfaces were comparatively smooth, while the affinity for laminin-111 was suitable for high-resolution HS-AFM imaging. Again, the domain organization of the short arms, as well as the central coiled-coil domain of the long arm and the LG domain cluster, could be observed well (Figure 1C), and only a few molecules were found in a condensed configuration with entangled short-arms. On bare mica, the laminin coiled-coil height values (2.2 ± 0.6 nm, mean ± S.D.) were in agreement with a previously reported value of 2.2 nm for the coiled-coil diameter obtained from EM micrographs [24,45], while the width values (4.8 ± 1.1 nm) were inflated due to tip-sample convolution effects (Appendix A). On APTES, the laminin coiled-coil height (1.9 ± 0.6 nm, mean ± S.D.) and width (3.4 ± 0.6 nm, mean ± S.D.) values were slightly reduced, potentially due to the partial embedding of the laminin molecule within the APTES layer. While the cross-shaped structure and domain organization could thus be observed using HS-AFM both on the bare mica surface at pH ≤ 5 and on the APTES-coated mica surface at pH 7.5 (Figure 1C, Appendix A), being able to image at physiological pH constitutes an important advantage of the APTES surface protocol. 

### 2.2. Investigating the Coiled-Coil Domain Structure of Laminin-111

The majority of laminin-111 molecules on the APTES-mica surface at physiological pH displayed the characteristic cross-shaped structure of laminin and an S-shaped long arm (coiled-coil domain), similar to those seen in previous electron microscopy images [24,45]. The considerable deviation of the end-to-end distance (46.4 ± 18.4 nm) of the long arm (measured from the short arm junction to the end of the beginning of the LG domain cluster) from its contour length (66.3 ± 18.2 nm) further illustrates the bendiness of this domain (Figure 2B–D). Structural fluctuations primarily occurred in the laminin short arms, while the S-shaped coiled-coil domain of laminin-111 remained stationary in most molecules (Appendix A). Likewise, the radius distributions of the two opposing bends (bend 1: proximal to the LG domain end; bend 2: proximal to the short arm junction) were narrow and showed similar values of 9.2 ± 4.4 nm and 9.7 ± 5.9 nm (Figure 2E–G), indicating that the S-shape of the laminin-111 coiled-coil domain is a structurally stable configuration.

### 2.3. HS-AFM Imaging of Laminin-3A32 and Laminin-3B32

After having established suitable HS-AFM imaging conditions for laminin-111, it was tested if these conditions could also be employed for the investigation of other laminin family members. We extended our HS-AFM analysis to laminin-332 (previously laminin-5), using human recombinant protein produced in HEK293 cells [46]. In contrast to the α1-chain of laminin-111, which carries five C-terminal LG domains, the α3-chain product of recombinant laminin-332 contains only the first set (LG1-3) of the LG domain tandem as a result of post-translational processing [27]. Furthermore, there are two splice variants of the laminin α3-chain (α3A and α3B, to form laminin-3A32 and laminin-3B32, respectively), which differ in the length of the short arm [47,48]. Laminin-332 has a predicted pI of 6.4, which is closer to physiological pH than the predicted pI of laminin-111 (5.3). Accordingly, both laminin-332 isoforms adsorbed well to bare mica and could be imaged with high fidelity even in pH 7.5 imaging buffer (Figure 3 and Appendix A). Likewise, both isoforms also yielded high-quality images when immobilized and imaged on 0.01% APTES-coated mica (Figure 3 and Appendix A). The resolution of the HS-AFM images was also sufficient to reveal structural differences between laminin-3A32 and laminin-3B32. The α3A, β3, and γ2 chains of laminin 3A32 lack some or all the globular N-terminal globular domains present in other laminin isoforms and therefore feature truncated short arms (Figure 4A). The three truncated short arms of laminin-3A32 typically formed a condensed, globular structure (Figure 4B) in which individual short arms could not be distinguished, while a larger globular feature at the opposite end represented the tightly clustered LG1-3 domains. In contrast, the laminin-3B32 isoform contains a full length α3-chain (Figure 4C) and individual short arms carrying globular domains could be distinguished in some AFM frames (Figure 4D).

Importantly, the possibility to image both laminin-111 and the two laminin-332 isoforms on the same substrate and in the same pH 7.5 imaging buffer provided an opportunity to compare dynamic conformational changes between both isoforms. In contrast to laminin-111 molecules, the majority of which featured stationary, curved yet inflexible coiled-coiled domains, the coiled-coil domain of both laminin-332 isoforms displayed great flexibility, frequently switching between straight and bent conformations around an apparent hinge region (Figure 4B,D, Appendix A). In the bent conformation, the N-terminal and C-terminal segments of the coiled-coil domain typically remained primarily straight, but sharply changed direction at the hinge situated roughly in the middle of the coiled-coil domain. Tracing the angle between the coiled-coil segments at the hinge position for three individual laminin-3A32 molecules over time showed perpetual small angle fluctuations, interspersed with occasional sudden large angle changes (Figure 4E), with a mean angle of 63.0 degrees (Figure 4F). While it had been previously suggested that the laminin-332 coiled-coil likely undergoes reversible dynamic transitions around a central hinge region based on the simultaneous presence of molecules in different bending angles in EM micrographs, HS-AFM can visualize such dynamic transitions in real-time.

### 2.4. Identifying the Hinge Region in the Laminin-332 Coiled-Coil Domain

To narrow down the exact position of the hinge region, we traced the entire visible contour lengths of the full coiled-coil domain (67 ± 9.8 nm, Figure 4G), as well as the position of the hinge from the end of the C-terminus of the coiled-coil domain (24.0 ± 4.1 nm, Figure 4H) in laminin-3A32. For this, we determined the distance from the coiled-coil proximal edge of the LG1-3 domain cluster to the short arm junction, or the kink position. However, available crystal structures of the C-terminus of the coiled-coil and the adjacent LG1-3 domains of laminin-111 [33], and laminin-511 [34] demonstrate that the C-terminal coiled-coil domain extends to the basal side of the LG1-3 domains. The very C-terminus of the coiled-coil domain is therefore obscured in AFM topographies by the overlying LG1-3 domains. Assuming an AFM tip radius of 2 nm and accounting for lateral feature widening due to tip-sample convolution, we determined that the LG1-3 domain cluster obscures ~9 nm of the coiled-coil domain in AFM images (Appendix A). For determining the true hinge distance from the C-terminus of the coiled-coil domain, we therefore added this value to the average kink position distance from LG1-3, yielding a combined distance of 24.0 + 9.0 nm = 33.0 nm. The exact ratio of amino acids per nm of triple coiled-coil is unknown for laminin-332. However, from the crystal structure of a truncated laminin-511 E8 fragment, we determined a rise of ~0.15 nm per amino acid. Applying the same conversion factor to the laminin-332 coiled-coil yielded 33.0 nm/0.15 nm × AA − 1 = 220 AA as the most probable distance from the C-terminus of the coiled-coil, identifying a region near AA 2169 of the α3 chain. To support this finding obtained from AFM image analysis, we employed a complementary structural data prediction approach based on AlphaFold machine learning [49]. First, we verified if AF could reliably predict a laminin triple coiled-coil structure based on amino acid sequence input. For this, we compared the AF-predicted conformation of the C-terminal coiled-coil of laminin-511 to the corresponding known crystal structure of this fragment (PDB 5XAU). The AlphaFold-predicted structure showed excellent agreement with the corresponding coiled-coil segment of the crystal structure (root mean square deviation of alpha-carbon atom positions: 1.7 Å, Figure 5), demonstrating that AlphaFold can predict laminin triple coiled-coil structure accurately. Next, we applied this approach to predict the structure of the trimeric coiled-coil region of laminin-332 over a 200 AA stretch containing the putative kink region seen in the AFM images. AlphaFold predicted both straight and kinked structures for this fragment (Figure 6A), closely resembling the conformations observed by experimental AFM imaging. The predicted kinked structures display a partial α-helix unwinding of the laminin α-, β-, and γ-chains within the hinge, likely contributing significantly to the structural flexibility of this region, as well as AA side chain rotation away from the coiled-coil center. Lastly, the conformational dynamics of the laminin-332 coiled-coil structure predicted by AlphaFold were explored by performing coarse-grained molecular dynamics simulations using CafeMol [50] (Figure 6B). These simulations demonstrated repeated cycling between bent and straightened conformations, again closely resembling the experimental HS-AFM results. Combining experimental HS-AFM imaging with AlphaFold-based structure prediction and molecular dynamics simulation thus generated matching complementary structural information pinpointing the position of the flexible hinge region with the coiled-coil domain of laminin-332.

### 2.5. Visualizing the Dynamic Rearrangement of Laminin LG Domains Using HS-AFM

In addition to visualizing coiled-coiled dynamics, the generated HS-AFM timelapse series also detected reversible conformational changes in the LG domains at the C-terminus of the α1-chain of laminin-111 (Figure 7A, Appendix A) and the α3-chain of laminin-3B32 (Figure 7B, Appendix A). In some movies of laminin-111, the LG1-5 domains could be observed changing from a compact into an open conformation, in which the LG domains are more loosely associated (Figure 7A). However, even in the open arrangement, the LG1-5 domains of laminin-111 remained still close to each other, and individual domains could not be clearly distinguished in most HS-AFM movies. Laminin-3B32 lacks the LG4-5 domains due to proteolytic processing. The HS-AFM observations showed the LG1-3 domains in various conformations, including “compact”, “open”, and “stretched” (Figure 7B). Different open and closed conformations of LG domains from different laminin isoforms have been previously observed using electron microscopy [45] and X-ray crystallography [53], and could play a role in regulating integrin receptor binding to the LG domain region.

## 3. Discussion

In this study we established suitable surface immobilization and imaging conditions to visualize dynamic conformational changes within different laminin isoforms using HS-AFM. While imaging on bare mica surfaces required optimized buffer conditions (pH ≤ 5) for the stable imaging of laminin-111, APTES-modified mica surfaces could be used for comparative laminin-111 and laminin-332 imaging at physiological pH (7.5). The obtained HS-AFM movies revealed the dynamic rearrangement of the laminin-332 coiled-coiled domain, cycling between extended and kinked conformations around a defined hinge region, while the laminin-111 coiled-coil displayed little flexibility and remained in a stable, S-shaped conformation previously observed using electron microscopy [24]. Furthermore, by complimenting experimental HS-AFM scanning with simulated AFM image reconstruction, AlphaFold-based structural prediction, and molecular dynamics simulation, a putative hinge location could be identified located ~220 AA upstream of the coiled-coil C-terminus of laminin-332 (α3: Glu^2167^-Ser^2164^, ß3: Val^945^-Arg^941^, γ2: Asp^963^-Val^962^). Based on AlphaFold structure prediction, during kinking the hinge region displays a partial unwinding of the α3-, ß3-, and γ2-chains, which may provide a structural mechanism for the hinge flexibility. Furthermore, AA side chains in the hinge regions primarily point away from the coiled-coil rod in the hinge regions, which may further aid coiled-coil kinking. Performing coarse-grained molecular dynamic simulations on the kinked coiled-coil structure predicted by AlphaFold confirmed the high flexibility of the hinge region and yielded structure snapshots closely resembling the experimental AFM images. While experimental AFM data and structure predictions and simulations thus agreed remarkably well in pinpointing the location of the hinge and in kink angle prediction, it should be noted that molecular dynamics simulations cannot yet capture the full structural dynamics of laminin 332, including the slow complete transitions between straight and kinked shapes as seen in HS-AFM movies. AFM imaging requires the surface attachment of the sample, which may influence transition speed and frequency. In agreement, increased laminin-332 flexibility on bare mica compared to APTES-mica surface (Appendix A) suggests a notable contribution of the sample substrate on coiled-coil dynamics during HS-AFM observations.

So far, the functional relevance of the structurally more stable S-shape of the laminin-111 coiled-coil domain, or the dynamic kinking of the laminin-332 coiled-coil are unknown. The cross-shaped laminin ultrastructure likely reflects its role as a multi-adapter protein in the extracellular matrix and a cell adhesion promoter. Flexible short arms may facilitate the formation of the ternary laminin node between α-, β-, and γ-chain LN domains [16,54] and provide spatially adjustable interaction points with collagen networks within the basement membrane, while the elongated coiled-coil domain of the laminin long arm could help to present the integrin- and dystroglycan-binding sites to epithelial or endothelial layers at the top of the basement membrane. While coiled-coil flexibility may increase the probability of successful engagement of integrin-binding sites located at the coiled-coil/LG1-3 interphase [34,55], establishing if such mechanisms are relevant for laminin-332 function requires further investigation. Alternatively, kink regions could contain enzymatic cleavage sites, as suggested previously [56], or binding sites for other matrix proteins or cellular receptors, access to which could be regulated by the degree of coiled-coil bending. For instance, the binding of the proteoglycan agrin to the coiled-coil domain of γ1-containing laminin isoforms is controlled by coiled-coil conformation [57]. A fraction of laminin-332 molecules displayed a near-complete back-bending (~0 degree angle) of the C-terminal half of the coiled-coil domain (Appendix A), bringing the LG1-3 domains in close proximity to the laminin short arms and suggesting the possibility of intramolecular interactions between short arm and LG domains.

While coiled-coil domains are typically stable structural motifs facilitating firm protein association [58], coiled-coil domains of many proteins, including myosins [59], contain spatially defined flexible regions permitting dynamic motion essential for carrying out protein-specific functions [60]. These flexible regions may be short and are often characterized by an interruption of the regular heptad repeat pattern within the coiled-coil [61]. Combining machine learning-based structure predictions and molecular dynamics simulations with complementary high-resolution AFM imaging provides exciting new possibilities for elucidating molecular mechanisms governing dynamic coiled-coil kinking in laminins and other proteins.

HS-AFM furthermore revealed the dynamic cycling of the LG1-5 domains of laminin-111 and the LG1-3 domains of laminin-332 domains between compact and loose arrangements. While individual LG domains could not be clearly resolved in laminin-111 images even in the “open” conformation, the three LG domains of laminin-332 could be frequently observed to assume an “open” conformation characterized by clearly spaced-apart individual LG domains. Previous crystallographic and cross-linking studies have identified an essential role of the γ-chain tail in organizing the LG1-3 domains from an “open” confirmation, in which the LG1 domain is dissociated from the LG2-3 pair [53], into an compact or “cloverleaf” configuration with full integrin-binding activity [33,34,55,62]. However, the HS-AFM experiments presented here demonstrate conformational cycling between “open” and “closed” conformations in the presence of the γ-chain, indicating that under physiological conditions, the integrin-binding interface may be structurally more dynamic than crystallographic studies suggest. In conclusion, our findings underline the power of HS-AFM analysis for observing dynamic conformational changes in proteins. In future, it will be interesting to characterize dynamic changes in additional laminin isoforms using HS-AFM.

## 4. Materials and Methods

### 4.1. Sample Preparation

Stock solutions (100 µg/mL) of human recombinant laminin-111 (LN111-02) and laminin-3B32 (LN332-0202) were purchased from Biolamina (biolamina.com). Human recombinant laminin-3A32 (47200000, 20 µg/mL) was purchased from Oriental Yeast Co. Ltd. Nagano, Japan. Stock solutions were aliquoted and stored at −80 °C and diluted with the imaging buffer immediately before deposition on mica for HS-AFM imaging.

### 4.2. Cantilever Preparation

BL-AC10DS-A2 (Olympus Micro Cantilevers, Shinjuko City, Japan) cantilevers with a spring constant of 0.1 N/m and a resonance frequency of 0.4 MHz in water (1.5 MHz in air) were used as the scanning probe to visualize the samples using HS-AFM. The dimensions of the cantilevers were as follows: 9 µm (length), 2 µm (width), and 0.13 µm (thickness). Electron beam deposition (EBD) was performed to enhance the HS-AFM imaging resolution by fabricating an amorphous carbon tip on the original tip of the cantilever using a scanning electron microscope (ELS-7500, Elionix, Tokyo, Japan). The length of the additional AFM tip was ∼300–500 nm, and typical tip apex radii ranged between ∼1 and 4 nm. Tips were further sharpened using plasma etching.

### 4.3. HS-AFM Observation

A laboratory-built HS-AFM with a narrow-range scanner was used to visualize the proteins in the tapping mode as previously described [63]. A laser beam (670 nm) was focused on the cantilever tip through a 20× objective lens to detect the cantilever deflection. The free oscillation amplitude of the cantilever was 2 nm, and the set-point amplitude for the feedback control was 80–90% of the free amplitude. To minimize the tip-sample interaction force, the feedback parameters were optimized during scanning. A sample glass stage (2 mm diameter, 2 mm height) carried a muscovite mica disc (0.1 mm thickness, 2 mm diameter) on top, attached with cyanoacrylate glue. Just before deposition on the freshly cleaved mica substrate, samples were diluted to a final concentration of 1 µg/mL in the respective imaging buffer (150 mM NaCl, 50 mM glycine-HCl, 1 mM CaCl_2_, pH 3.5; 150 mM NaCl, 50 mM MES, 1 mM CaCl_2_, pH 5; 150 mM NaCl, 50 mM MES, 1 mM CaCl_2_, pH 6; 150 mM NaCl, 50 mM Tris, 1 mM CaCl_2_, pH 7.5; 150 mM NaCl, 50 mM glycine-NaOH, 1 mM CaCl_2_, pH 9; 150 mM NaCl, 50 mM glycine-NaOH, 1 mM CaCl_2_, pH 10). After 10 min of incubation, substrates were rinsed 5x with imaging buffer to remove unabsorbed proteins and subsequently scanned using HS-AFM. To observe proteins on (3-aminopropyl) triethoxysilane (APTES) surfaces, freshly cleaved mica was treated with 1%, 0.1%, or 0.01% APTES dilutions in water for 3 min and subsequently washed 5× with 5 µL of water and 5x in imaging buffer (150 mM NaCl, 50 mM Tris, 1 mM CaCl_2_, and pH 7.5). Afterwards, protein samples (2 µL) were deposited in imaging buffer, incubated for 10 min, and then rinsed again in imaging buffer (5×) before image acquisition. All HS-AFM experiments were performed at room temperature. The isoelectric point (pI) of laminin isoforms were predicted using the Prot pi Protein Tool (www.protpi.ch/, accessed on 1 February 2024)

### 4.4. HS-AFM Image Processing

HS-AFM raw images were processed using the Gwyddion 2.60 software (gwyddion.net) by applying mean plane subtraction, row alignment, and the correction of horizontal scan lines (if needed). Height scale bars were also prpeared in Gwyddion 2.60. HS-AFM movies were prepared using in-house software routines developed in MATLAB (Mathworks, Natick, MA, USA, R2021) [64]. Movies were flattened by applying a plane or second-order polynomial surface fitting, as appropriate. Image frame drift was corrected using a 2D alignment feature of the Matlab module. Timepoint and scale bars were added using image J 1.53v (www.imagej.nih.gov/ij/, accessed on 1 February 2024).

### 4.5. Data Analysis

Laminin coiled-coil length and kink angle were measured using Image J. For determining the radius of bends, circles were fitted on curvatures of the laminin-111 coiled-coil manually. Histograms and Gaussian fittings were prepared in MATLAB. For time traces, angles around the coiled-coil kink position were measured using Image J, and values were plotted in Origin Pro 2021 (www.originlab.com, accessed on 1 February 2024). 

### 4.6. Structure Prediction of Laminin Coiled-Coiled Region

To support the interpretation of the HS-AFM imaging of laminin conformational dynamics, we employed structural data predicted using the machine learning AlphaFold v2 approach based on amino acid sequence information [49]. For convenient implementation, we used the free ColabFold platform [65]. First, we verified the method accuracy by comparing the AlphaFold-predicted coiled-coil conformation with that of the known crystal structure of laminin-511 (PDB 5XAU). The prediction was performed for the following amino acid sequences: alpha5 Val^2676^–Val^2735^; beta1 Arg^1716^–Leu^1786^; gamma1 Leu^1535^–Pro^1609^. Structure similarity was quantified by determining the root mean square deviation (RMSD) of alpha-carbon atom positions for the set of amino acid residues shared between the crystal and the AlphaFold-predicted structure for the coiled-coil segment. The obtained RMSD was 1.7 Å. We then proceeded with the structure prediction for the laminin-332 trimeric coiled-coil structure containing the putative kink region seen in HS-AFM images. Here, we used the following sequences: α3 Leu^2100^–Asp^2300^; β3 Ser^877^–Gly^1077^; γ2 Lys^893^–Val^1093^ from the UniProt LAMA3 (Q16787), LAMB3 (Q13751), and LAMC2 (Q13753) sequences.

### 4.7. Simulated AFM Images of Laminin Structures

We employed the simulation AFM of available structural data (PDB 5XAU) and of the AlphaFold-predicted structures to compute pseudo AFM images that can be compared with measured AFM images. The BioAFMviewer 3.0 software (https://www.bioafmviewer.com/, accessed on 1 February 2024) [66] was used to simulate scanning based on the non-elastic collisions of a rigid cone-shaped tip model with the rigid van der Waals atomistic structure. A scan step of 1 nm was used, and the tip shape parameters were 1 nm for the tip probe sphere radius and 5° for the cone half-angle. 

### 4.8. Molecular Dynamics Simulation of Structural Dynamics

To explore the conformational dynamics of the laminin-332 coiled-coil structure predicted using AF, we performed coarse-grained molecular dynamics simulations employing the CafeMol simulation package [50]. The Langevin dynamics approach with a flexible local potential for intra-chain interactions and the Go potential, excluded volume repulsion, electrostatic, and hydrophobic contributions for non-local interactions was used. We refer to the CafeMol manual [50] for a detailed description of the model.

## Figures and Tables

**Figure 1 ijms-25-01951-f001:**
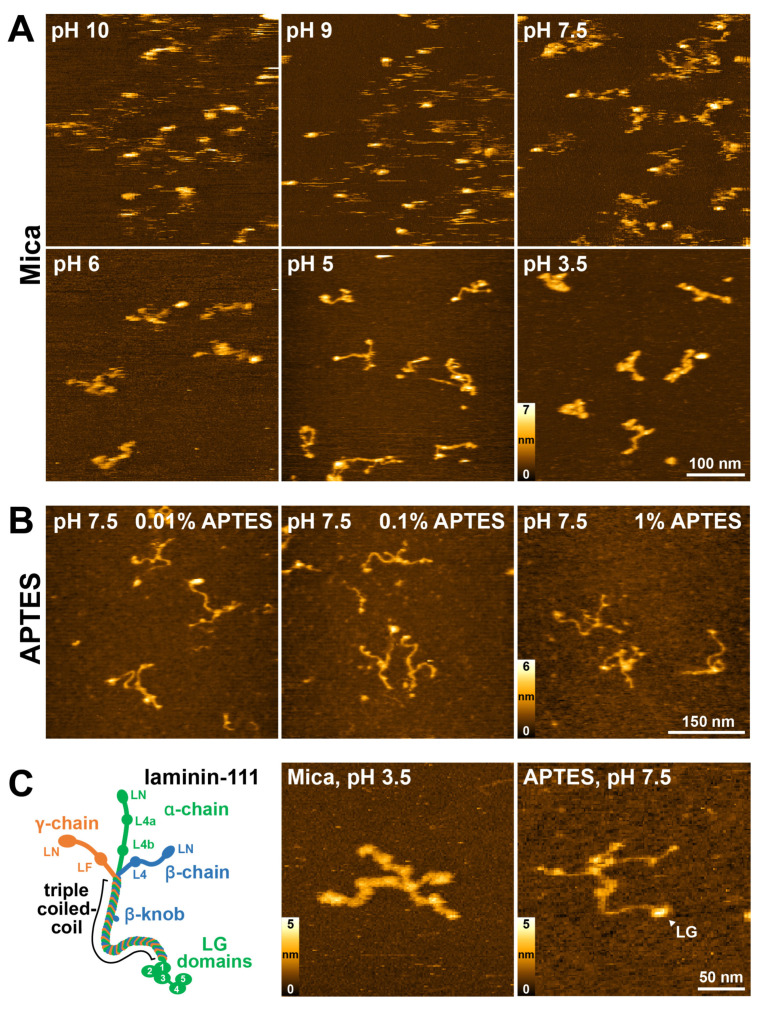
Surface immobilization optimization for the high-resolution imaging of laminin-111 using HS-AFM. (**A**) HS-AFM images of laminin-111 on bare mica surfaces immobilized and imaged in solutions with decreasing pH values. (**B**) HS-AFM images of laminin-111 immobilized on mica coated with different concentrations of APTES and imaged in pH 7.5 imaging solution. (**C**) Schematic diagram of laminin-111 formed through association of α- (green), β- (blue), and γ- (orange) chains. Numbers denote the five C-terminal LG domains of the α-chain (**left image**). HS-AFM images of laminin-111 on a bare mica surface at pH 3.5 (**middle image**) and an 0.1% APTES-coated mica surface at pH 7.5 (**right image**). LG—Laminin globular domain indicated by an arrow. Scale bar 50 nm.

**Figure 2 ijms-25-01951-f002:**
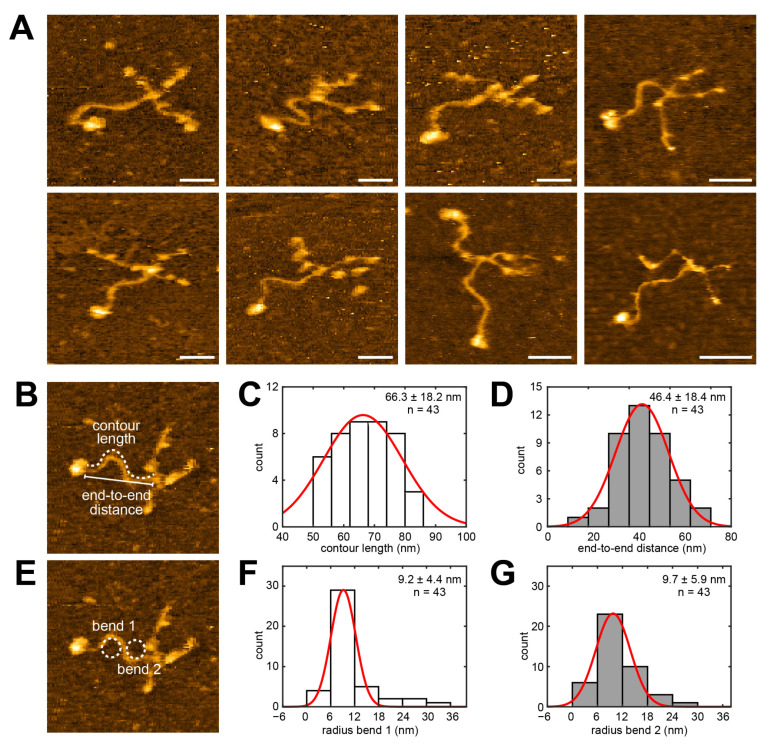
Analyzing the coiled-coil structure of laminin-111 using HS-AFM. (**A**) Panel of HS-AFM images of laminin-111 in different conformations. Scale bars 30 nm. (**B**) The contour length and end-to-end distance of laminin-111 coiled-coil are indicated by a dashed line and solid line, respectively. Image size 160 nm × 160 nm. (**C**) Histogram and Gaussian fit (red curve) of contour length of laminin-111. (**D**) Histogram and Gaussian fit (red curve) of the end-to-end distance of the coiled-coil of laminin-111. (**E**) Two dashed line circles fitted in the coiled-coil curvature of laminin-111. Image size 160 nm × 160 nm. (**F**) Histogram and Gaussian fit (red curve) of the radius of the fitted circle to the first bend of the laminin coiled-coil seen from the LG domain. (**G**) Histogram and Gaussian fit (red curve) of the radius of the fitted circle to the second bend of the laminin coiled-coil seen from the LG domain. n denotes the number of molecules analyzed.

**Figure 3 ijms-25-01951-f003:**
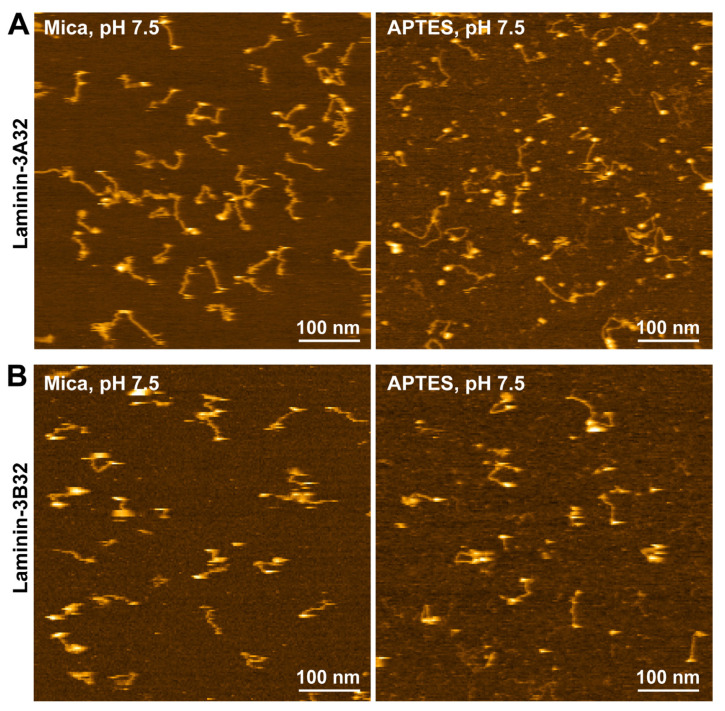
HS-AFM images of laminin-3A32 and -3B32 isoforms on mica and APTES surfaces at pH 7.5. (**A**) HS-AFM images of laminin-3A32 on a mica (**left**) and APTES substrate (**right**). (**B**) HS-AFM images of laminin-3B32 on a mica (**left**) and APTES substrate (**right**) at pH 7.5.

**Figure 4 ijms-25-01951-f004:**
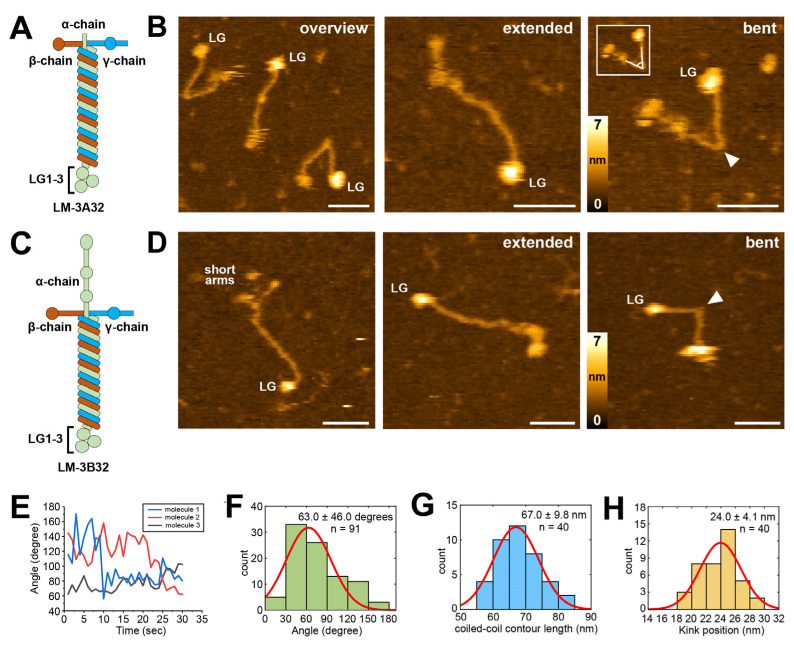
Observation of laminin-3A32 and -3B32 using HS-AFM. (**A**) Schematic depiction of laminin-3A32 consisting of α (light green), β (light blue), and γ (orange) chains. (**B**) Overview HS-AFM image of LM-3A32 in different conformations (**left**). High magnification images showing the laminin coiled-coil in extended (**middle**) and bent (**right**) conformations. The kink position is indicated by the white arrowhead. A cropped insert framed by a white box illustrates how the kink angle was measured in (**F**). Scale bars 30 nm. (**C**) Schematic depiction of laminin-3B32. (**D**) HS-AFM images of laminin-3B32 with separated short arms (**left**) and in extended (**middle**) and bent (**right**) coiled-coil conformations. The kink position is indicated by the white arrowhead. Scale bars 30 nm. (**E**) Time traces of angle change of three individual laminin-3A32 molecules. (**F**) Histogram fit of the angle of the coiled-coil kink. (**G**) Histogram fit of laminin-3A32 coil-coil length. (**H**) Histogram fit of the length from the starting point of the coiled-coil from the LG domain to the kink position. n is the number of molecules analyzed for each histogram. Red curves indicate Gaussian fits.

**Figure 5 ijms-25-01951-f005:**
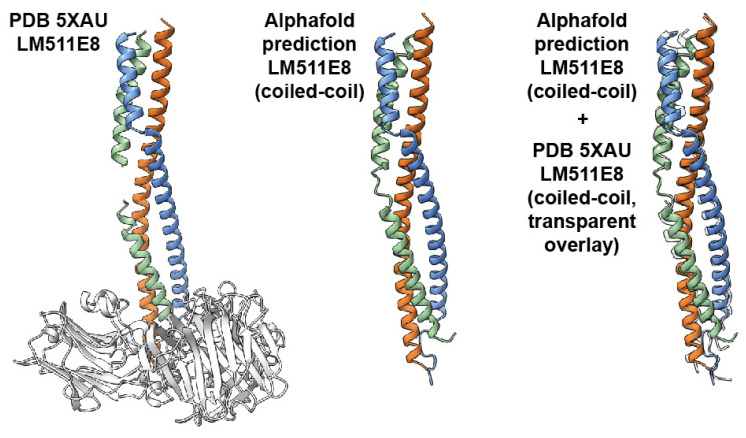
(**Left**): crystal structure of the integrin-binding fragment of laminin 511 (LM511E8). The α-, β-, and γ-chains are colored in green, red, and blue, respectively. LG1-3 domains are shown in gray. (**Middle**): structure of the α, β, and γ coiled-coil segment of LM511E8 predicted by AlphaFold. (**Right**): structure of the α, β, and γ coiled-coil predicted by AlphaFold (solid colors) superimposed with the corresponding part of the 5XAU crystal structure (transparent, RMSD 1.7 Å).

**Figure 6 ijms-25-01951-f006:**
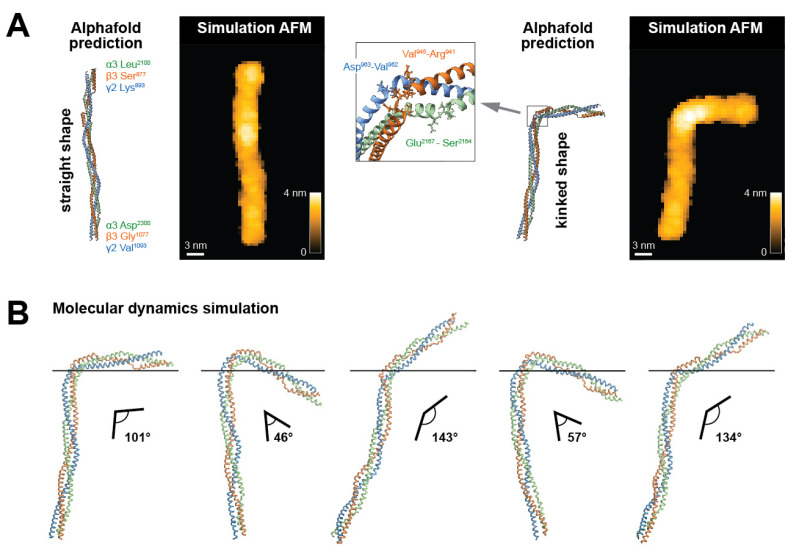
(**A**) Atomistic structure of the coiled-coil part of laminin 332 containing the putative kink as predicted by AlphaFold. The α-, β-, and γ-chains are colored in green, red, and blue, respectively. The straight shape (**left**) and kinked shape structure (**right**) are shown together with their corresponding simulated AFM images. For the kinked shape conformation, the hinge region is magnified and the side-chains corresponding to unstructured parts within the chains are shown. The straight shape of the coiled-coil structure corresponded to the AlphaFold “rank 1” prediction, and that of the kinked shape was the “rank 4” prediction. (**B**) Snapshots obtained from coarse-grained molecular dynamics simulations of the AlphaFold kinked shape structure, confirming the flexibility of the hinge region. ChimeraX (https://www.cgl.ucsf.edu/chimerax/, accessed on 1 February 2024) [51] and VMD (https://www.ks.uiuc.edu/Research/vmd/, accessed on 1 February 2024) [52] were used for the visualization of the molecular structures.

**Figure 7 ijms-25-01951-f007:**
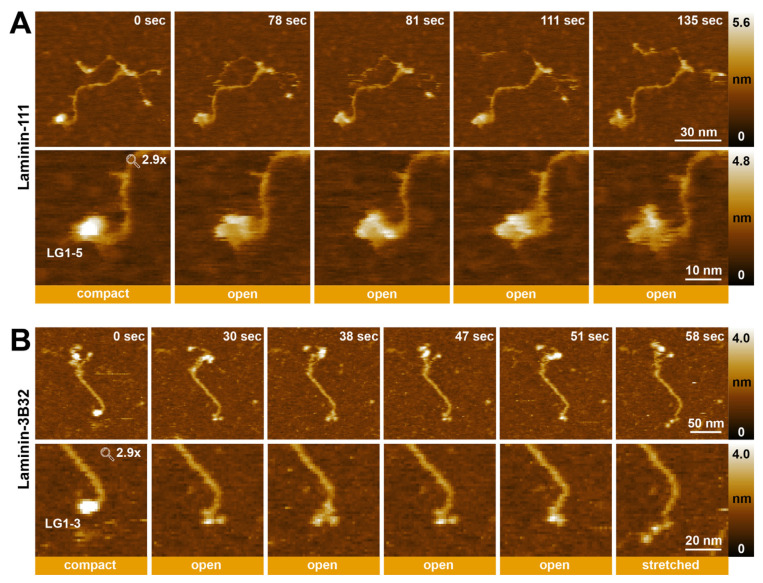
Conformational changes of laminin LG domains observed using HS-AFM. (**A**) Successive HS-AFM images of laminin-111 recorded at 3 frames per second (**upper row**). Cropped images of the images showing conformational changes of the LG1-5 domains (**lower row**). (**B**) Successive HS-AFM images of full-length laminin-3B32 recorded at 3 frames per second (**upper row**). Cropped images of the upper row images showing conformational changes of LG1-3 domains (**lower row**).

## Data Availability

The original contributions presented in the study are included in the article/Appendix A; further inquiries can be directed to the corresponding authors.

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
