# Peer review of "Observing Dynamic Conformational Changes within the Coiled-Coil Domain of Different Laminin Isoforms Using High-Speed Atomic Force Microscopy"

_ijms, 2024, doi:10.3390/ijms25041951_

Round 1
Reviewer 1 Report
Comments and Suggestions for Authors
In this work, Akter et al. present new data on the dynamic behavior of laminins (111 and 332) obtained through high-speed AFM. They initially assess various adsorption conditions of the sample to the mica surface, discovering significant differences in protein dynamics linked to buffer pH. Upon selecting conditions closer to physiological pH, they investigate the dynamics of laminins (111 and 332) coiled-coil and LG domains. They observe that while the coiled-coil of laminin 111 maintains an S-shape and is less flexible, in laminin 332, there is a kink region exhibiting high dynamic behavior. In addition, they identify fluctuations in the LG domains, which transition between open, compact and stretched conformations. Overall, I find that the manuscript is well-presented, with clear analysis, controls, and figures. I believe it is suitable for publication with minor revision.
- Some videos are very short and it is difficult to appreciate the features you describe. Particularly, videos S4 and S5 are only 1 sec long and videos S2 and S6 are only 2 seconds long. Please, provide videos of at least 3-4 seconds.
- Sometimes authors distinguish between laminin 3A32 and 3B32, and sometimes they refer 332, which become confusing. They should refer to each one along the whole manuscript and avoid the generic reference 322,
- Line 151, pH 7.5 instead of 7.6
- Line 247, You mention “… 3A32 molecules over time showed perpetual small angle fluctuations …”. For how long have you measured these fluctuations? You mention that mean angle is 63º but, which is the range of fluctuation?
- Line 269, Fig. 4F instead of Fig. 4E
- Line 271, Fig. 4H instead of Fig. 4G
- Line 293, root mean square deviation 1.7. Å. Is this a superposition or a structural alignment? I guess this is only using C alphas, isn’t it? over how many residues is the superposition/alignment?
- Line 304, Fig. 6B instead of Fig. 7B
- I would avoid the simplification of AlphaFold2 into AF (it is very similar to AFM and could induce missunderstandings)
- Line 313, AlphaFold 2 instead of AlphaFold
- Figure 5. Please, keep the color scheme on the left and the right-hand sides (i.e., the crystal structure showing each chain in one color). Then, color AlphaFold2 prediction in a different color (use a darker color because transparent is very difficult to see).
- Figure 6B, please label the angles of each molecule bend around the kink region
- Line328, Fig. 7A instead of Fig. 8A
- Line 331, Fig. 7A instead of Fig. 8A
- Figure 7. You have label panel A and C but miss the labeling of panels B and D. Please label which is the open and which is the compact conformation of LG domains in B. Please label which is the open, which is the compact and, which is the stretched conformation in panel D. In line 343 you mention laminin 3B32, whereas in the panel it is Laminin 332. Line 344 “… images shown in (a) showing …” I guess you mean “shown in (c)”.
- Line 391. Please label in video S5 with arrows molecules that display near complete back-bending
- Line 446. 50 mM MES, 1 mM CaCl2, pH 5. MES buffering range is from 5.5 to 6.7, is you buffer pH 5.5?
- Line 449, “ … substrates were rinse 5x with imaging buffer…”, which volume is this?
- Line 451. Upon APTES treatment, how did you incubated and prepare protein sample?
- Figure S2. Please label in Å the width of the LG1-3 domain and the height of the coiled-coil in the crystal structure (also in B)
- There is no reference to video S9 within the text.
Reviewer 2 Report
Comments and Suggestions for Authors
Round 2
Reviewer 1 Report
Comments and Suggestions for Authors
I have only detected two issues in movies S5 and S6, where the righthand side of the image becomes black from second 3 to the end. It could be a problem with the software on my computer or a real issue within the movies.
I cannot see the circle indicating a molecule displaying backbending in movie S5 (but this could be because of the problem mention above).
Reviewer 2 Report
Comments and Suggestions for Authors
The authors have addressed my concerns so I consider the paper can be published.